# Effect of the COVID-19 pandemic on the care for acute cholecystitis: a Swedish multicentre retrospective cohort study

Erik Osterman  ,[1,2] Sofia Jakobsson,[3] Christina Larsson,[2] Fredrik Linder[3,4]

EO and SJ contributed equally.

¹CKF Gävleborg, Uppsala University, Gävle, Sweden
²Department of Surgery, Gävle Sjukhus, Gävle, Sweden
³Department of Surgical Sciences, Uppsala University, Uppsala, Sweden
⁴Department of Surgery, Uppsala University Hospital, Uppsala, Sweden

**Correspondence to**
Dr Erik Osterman;
erik.osterman@igp.uu.se

## ABSTRACT

**Objectives** The present study aimed to investigate if and how the panorama of acute cholecystitis changed in 2020 in Sweden. Seven aspects were identified, the incidence of cholecystitis, the Tokyo grade, the timing of diagnosis and treatment, the proportion treated with early surgery, the proportion of patients treated with delayed surgery, and new complications from gallstones.

**Design** Retrospective multicentre cohort study.

**Setting** 3 hospitals in Sweden, covering 675 000 inhabitants.

**Participants** 1634 patients with cholecystitis.

**Outcomes** The incidence, treatment choice and diagnostic and treatment delay were investigated by comparing prepandemic and pandemic patients.

**Results** Patients diagnosed with cholecystitis during the pandemic were more comorbid (American Society of Anesthesiologists 2–5, 86% vs 81%, p=0.01) and more often had a diagnostic CT (67% vs 59%, p=0.01). There were variations in the number of patients corresponding with the pandemic waves, but there was no overall increase in the number of patients with cholecystitis (78 vs 76 cases/100 000 inhabitants, p=0.7) or the proportion of patients treated with surgery during the pandemic (50% vs 50%, p=0.4). There was no increase in time to admission from symptoms (both median 1 day, p=0.7), or surgery from admission (both median 1 day, p=0.9). The proportion of grades 2–3 cholecystitis was not higher during the pandemic (46% vs 44%, p=0.9). The median time to elective surgery increased (184 days vs 130 days, p=0.04), but there was no increase in new gallstone complications (35% vs 39%, p=0.3).

**Conclusion** Emergency surgery for cholecystitis was not impacted by the pandemic in Sweden. Patients were more comorbid but did not have more severe cholecystitis nor was there a delay in seeking care. Fewer patients non-operatively managed had elective surgery within 6 months of their initial diagnosis but there was no corresponding increase in gallstone complications.

## BACKGROUND

Early cholecystectomy is the recommended treatment for acute cholecystitis. Non-operatively managed (NOM) patients may be considered for surgery later to avoid new complications from cholecystolithiasis.[1] During the COVID-19 pandemic, some elective surgical procedures were deferred to

## STRENGTHS AND LIMITATIONS OF THIS STUDY

⇒ Covers all patients seeking hospital care for cholecystitis in a developed country with few restrictions.
⇒ Uses both electronic patient records and prospectively collected registry data.
⇒ Any patient solely cared for in primary care not included.
⇒ Only includes 2020 as a pandemic year.
⇒ Mainly retrospective study.

handle the effects of the pandemic.[2] The pandemic affected uncomplicated cholecystectomies more than other procedures,[3 4] and the number of elective cholecystectomies in Sweden decreased by 30% while emergency procedures only decreased by 1%.[5] In 2020, the pandemic hit Sweden in two waves, the first starting in March and the second in October.[5] Sweden had less strict policies to stop the spread of the virus than many other countries.[6]

### Previous studies

Increased severity of cholecystitis and fewer patients being treated with early cholecystectomy for cholecystitis have been seen during the COVID-19 pandemic.[7 8] The Swedish Registry of Gallstone Surgery and Endoscopic Retrograde Cholangiopancreatography (Gallriks)[9] reported fewer elective procedures while emergency procedures increased, but no data were available on non-operative management.[10] The incidence of cholecystitis during the pandemic has not been completely investigated, cholecystitis decreased in the USA[11] while it remained the same in Canada,[12] increased in Germany[8] and the CHOLECOVID study[7] showed a decrease in mild cholecystitis but not moderate and severe cholecystitis. Emergency department visits fell in many places,[5 11 12] which could indicate that patients waited at home and then presented after 5–7 days when conservative treatment is recommended explaining falling emergency surgery numbers. However,

a German study investigated the proportion of non-surgically treated patients and found that non-operative management did not increase during the pandemic.[13]

## Aims and hypothesis

The present study aimed to investigate if and how the panorama of acute cholecystitis changed during 2020 at three Swedish centres in two counties. Seven aspects were identified: the incidence of cholecystitis, the Tokyo grade,[14] the timing of diagnosis and treatment, the proportion treated with early surgery, the proportion of patients treated with delayed surgery, and new complications from gallstones. The hypothesis was that the incidence of cholecystitis and the proportion treated with early surgery did not change while the grade increased, and patients were diagnosed later. We also hypothesised that fewer non-operatively treated patients had surgery within 6 months of discharge and more patients had new gallstone complications if diagnosed in 2020.

## METHOD

The manuscript was prepared following the Reporting of studies Conducted using Observational Routinely collected Data checklist.[15]

Patients with a diagnosis of acute cholecystitis between 2017 and 2020 were identified by the administrative data offices of Region Gävleborg and Region Uppsala. The data were then gathered from electronic patient records (EPR) and Gallriks as described previously.[16] The chart review started in mid-2021 and finished in early 2022. The possibility of chart review for the study staff determined the choice of regions.

The outcomes were: Incidence of cholecystitis (number of cases/100 000 inhabitants), grade (G1: mild or G2–3: moderate/severe),[14] the time to first contact with the emergency department and diagnosis from symptoms in days, time to surgery, treatment choice (surgery or NOM), time to delayed surgery, the proportion NOM patients who underwent surgery within 6 months and new gallstone complication after discharge. Exposure was a diagnosis of acute cholecystitis during the pandemic (after 13 March 2020).[17] Patients were stratified by diagnosis time, prepandemic or pandemic to analyse outcomes and demographics. To assess delays in delayed surgery, all patients diagnosed during 2020 were analysed as the exposed group. The number of inhabitants over 18 years of age in the two regions was obtained from Statistics Sweden and was transformed to person-years of exposure. The results were then checked against the national patient registry to further investigate the incidence of cholecystitis.

## Statistical analysis

Differences between groups were tested with the $\chi^2$ test for categorical variables and the Wilcoxon rank sum test for discrete variables. Post hoc testing with Holm correction for multiple testing was used for variables significant with more than two levels. The incidence was analysed in two ways, first calculated from the monthly rate and adjusted for the population size of the counties, and second by also adjusting for seasonality with the X11 procedure.[18] Logistic regression was used to analyse the proportion of NOM patients undergoing elective surgery within 6 months of their cholecystitis adjusting for comorbidities. A Cox proportional hazards model was used to investigate the time to new gallstone complications censoring for death and surgery. The number of patients each month and the proportion having surgery compared with the previous period were plotted. The monthly number of intensive care unit (ICU) admissions for COVID-19 was overlaid to convey the waves of the pandemic, the data came from the Swedish Intensive Care Registry's open reports.[19] Data were not imputed, and complete-case analysis was performed where applicable. Power analysis for an effect size of 0.3 with 95% power and α=0.05, for the Wilcoxon rank sum test and $\chi^2$ test with 4 df gave a required sample of at least 854 and 207 patients which determined the number of control years to include in the study. Logistic regression required a sample size of 636 to detect a 25% decrease in elective surgery within 6 months at 95% power and α=0.05. The Cox proportional hazards regression required a sample size of 954 to detect an increase of 50% (HR 1.5) in new complications at 95% power and α=0.05. Statistics were calculated with R V.4.2.2 (Vienna, Austria).

## Patient and public involvement

Patients or the public were not involved in the design, conduct, reporting or dissemination plans of our research.

## RESULTS

In total, 1634 patients were included in the study, 1286 patients were diagnosed before the pandemic and 348 after the start of the pandemic. The incidence of first-time cholecystitis did not change during the pandemic (76/100 000 cases/person-years prepandemic vs 81/100 000 cases/person-years, p=0.3, $\chi^2$ test, data in table 1). Adjusting for seasonality did not change the result. The national patient registry did not show a significant change in the incidence of cholecystitis between the years 2017–2019 and 2020–2021 either.

There was no difference in sex, age or body mass index between the two periods (table 2). During the pandemic, there was a 13% increase in patients classified as 2 in the American Society of Anesthesiologists physical classification system (ASA).[20] There was a corresponding 22% decrease in otherwise healthy patients (post hoc test ASA1 p=0.23 and ASA2 p=0.08). More patients had a CT as a diagnostic modality during the pandemic (59% prepandemic vs 67%, post hoc test p=0.007). Five patients had concurrent COVID-19.

**Table 1** Incidence of new cholecystitis 2017–2020, excluding recurrences during that period

| | Period | Person-years | Cases | Cases/100 000 | P value |
|---|---|---|---|---|---|
| Unadjusted | Prepandemic | 1 685 729 | 1286 | 76 | 0.3 |
| | Pandemic (From 13 March 2020) | 429 600 | 348 | 81 | |
| Seasonally adjusted | Prepandemic | 1 713 119 | 1297 | 76 | 0.7 |
| | Pandemic (From 1 April 2020) | 429 600 | 313 | 78 | |

P value calculated with $\chi^2$ test, only persons ≥18 years old used for person-years.

### Unchanged grade and admission times

There was no difference in the proportion of grade 1 vs grades 2–3 cholecystitis between the two periods (table 2). No difference in leucocyte counts and C reactive protein (CRP) at admission was noted suggesting that patients were not admitted later in the disease course (median leucocyte count $12.3 \times 10^9$/L prepandemic vs $12.3 \times 10^9$/L, p=0.53, median CRP 74 mg/L prepandemic vs 83 mg/L, p=0.62, Wilcoxon rank sum test). This was confirmed by a lack of differences in time to first contact with the emergency department and admission after symptoms. No difference in time to surgery from admission was noted either, suggesting that patients treated with surgery had the same priority as before the pandemic. There was no difference in the length of stay for patients treated with surgery (median 3 days, p=0.75) or without surgery (median 4 days, p=0.95).

### Treatment choice

There was no difference in the proportion of patients treated with early surgery during the pandemic (44% prepandemic vs 46%, p=0.41, table 2). There was no difference in the proportion of laparoscopic (38% prepandemic vs 38%), open (4% prepandemic vs 5%) or surgeries converted to open (2% prepandemic vs 3%) between the two periods (p=0.14).

For patients NOM, there was no difference in the proportion receiving drains or the plan for follow-up, 24% of patients were planned for delayed surgery either at discharge or after a visit to the outpatient clinic (p=0.24). Surprisingly, there was no difference in the proportion of patients referred to the outpatient clinic during the pandemic with 38% of NOM patients being referred and about 40% of those being planned for surgery during both periods (p=0.45).

### Temporal trends

The number of cases and the proportion treated with early surgery were calculated and compared with the 3 previous years to investigate if the pandemic changed the number of patients and the treatment choice month by month (figure 1). During the first 3 months of the year, there were approximately 20% more patients than the previous years and the proportion treated with early surgery was higher, in April, as the pandemic hit the total number of cases went down and a larger proportion of patients had surgery. During the second wave starting in October, the proportion treated with early surgery fell while cases remained comparable to 2017–2019 numbers. Changes were not statistically significant, and the figure is meant to illustrate the changes that were seen during the pandemic.

### Delayed surgery and new gallstone complications

The Tokyo guidelines consider delayed surgery to be surgery performed at least 42 days after acute cholecystitis.[1] The median time to surgery in the two regions in 2017–2019 was 130 days (IQR 76–214) while it was 184 days (IQR 78–341) in 2020 (p=0.04, Wilcoxon rank sum test). In 2017–2019, 20% of NOM patients had surgery within 6 months while in 2020 only 11% had surgery within this time (p=0.003, $\chi^2$ test). The only patient variable differing between the periods was ASA classification, and since this might affect the proportion treated with early surgery it was included in the logistic regression (table 3). Adjusted OR for surgery within 6 months in NOM patients was 0.47 (95% CI 0.29 to 0.74, p=0.002). This is in line with the longer median time to surgery seen in the unadjusted analysis.

The risk of new gallstone complications was not impacted by this, as 39% of patients NOM in 2017–2019 and 35% in 2020 had recurrent gallstone disease (p=0.3, $\chi^2$ test). There was no change in the pattern of new gallstone complications with new cholecystitis in 140 of 721 patients prepandemic (19%) and 33 of 188 (18%) patients during the pandemic. Cholangitis was the first complication in 14 patients (2%) prepandemic and 6 patients (3%) during the pandemic. Pancreatitis was the first complication in 11 patients (2%) prepandemic and 5 patients (3%) during the pandemic. The median time to a new gallstone complication was 87 days (IQR 29–249) prepandemic and 76 days (IQR 23–194) during the pandemic (n.s.). Accounting for the follow-up with a Cox proportional hazards model confirmed that there was no difference, HR 0.91 (95% CI 0.67 to 1.22, p=0.5).

### DISCUSSION

During the first month of the pandemic, there were fewer patients seeking hospital care for cholecystitis, which was compensated by more patients later in the year. Overall, there was no difference in the incidence of first-time

**Table 2** Demographics, treatment and outcomes for patients before and during the COVID-19 pandemic

| | Total | Prepandemic | Pandemic | P value |
|---|---|---|---|---|
| **Sex** | | | | |
| Male | 857 (52%) | 666 (52%) | 191 (55%) | 0.33 |
| Female | 777 (48%) | 620 (48%) | 157 (45%) | |
| **Age** | | | | |
| Median (IQR) | 67.2 (51.1–77.0) | 67.2 (50.7–76.8) | 67.2 (53.9–77.5) | 0.27 |
| **BMI** | | | | |
| Median (IQR) | 27.8 (24.7–31.4) | 27.8 (24.7–31.3) | 27.9 (24.7–31.5) | 0.98 |
| Missing | 117 (7.2%) | 93 (7.2%) | 24 (6.9%) | |
| **ASA** | | | | |
| 1 | 300 (18%) | 250 (19%) | 50 (14%) | 0.013 |
| 2 | 731 (45%) | 554 (43%) | 177 (51%) | |
| 3 | 515 (32%) | 405 (31%) | 110 (32%) | |
| 4 | 87 (5%) | 76 (6%) | 11 (3%) | |
| 5 | 1 (0%) | 1 (0%) | 0 (0%) | |
| **CCI** | | | | |
| Median (IQR) | 3.0 (1.0–5.0) | 3.0 (1.0–5.0) | 3.0 (1.0–5.0) | 0.43 |
| **Previous stone** | | | | |
| No | 1176 (72%) | 919 (71%) | 257 (74%) | 0.42 |
| Yes | 458 (28%) | 367 (29%) | 91 (26%) | |
| **Imaging** | | | | |
| No radiology | 59 (4%) | 50 (4%) | 9 (3%) | 0.01 |
| US | 579 (35%) | 475 (37%) | 104 (30%) | |
| CT | 570 (35%) | 424 (33%) | 146 (42%) | |
| CT+US | 424 (26%) | 336 (26%) | 88 (25%) | |
| Other+MRCP | 2 (0%) | 1 (0%) | 1 (0%) | |
| **Cholecystitis grade** | | | | |
| G1 | 798 (49%) | 627 (49%) | 171 (49%) | 0.85 |
| G2–3 | 729 (45%) | 569 (44%) | 160 (46%) | |
| Missing | 107 (7%) | 90 (7%) | 17 (5%) | |
| **Treatment** | | | | |
| Conservative | 815 (50%) | 642 (50%) | 173 (50%) | 0.41 |
| Surgery | 725 (44%) | 565 (44%) | 160 (46%) | |
| Drain | 94 (6%) | 79 (6%) | 15 (4%) | |
| **Time to first contact from symptoms** | | | | |
| Median (IQR) | 1.0 (0.0–3.0) | 1.0 (0.0–3.0) | 1.0 (0.0–3.0) | 0.78 |
| Missing | 57 (3.5%) | 45 (3.5%) | 12 (3.4%) | |
| **Time to admission from symptoms** | | | | |
| Median (IQR) | 2.0 (1.0–3.0) | 2.0 (1.0–3.0) | 2.0 (1.0–3.0) | 0.7 |
| Missing | 107 (6.5%) | 89 (6.9%) | 18 (5.2%) | |
| **Time to surgery from admission** | | | | |
| Median (IQR) | 1.0 (1.0–2.0) | 1.0 (1.0–2.0) | 1.0 (1.0–2.0) | 0.86 |
| **Length of stay (days)** | | | | |
| Median (IQR) | 3.0 (2.0–6.0) | 3.0 (2.0–6.0) | 3.0 (2.0–5.0) | 0.79 |
| Missing | 52 (3.2%) | 45 (3.5%) | 7 (2.0%) | |
| **Plan for follow-up** | | | | |

**Table 2** Continued

| | Total | Prepandemic | Pandemic | P value |
|---|---|---|---|---|
| No | 829 (73%) | 657 (72%) | 172 (76%) | 0.24 |
| Surgery | 304 (27%) | 251 (28%) | 53 (24%) | |
| Readmitted within 30 days | | | | |
| No | 1445 (88%) | 1138 (88%) | 307 (88%) | 0.92 |
| Yes | 189 (12%) | 148 (12%) | 41 (12%) | |

Fisher's exact test used for categorical variables, Wilcoxon rank sum test for discrete variables.
ASA, American Society of Anesthesiologists Performance Status Classification; BMI, body mass index; CCI, Charlson Comorbidity Index; MRCP, Magnetic resonance cholangiopancreatography; US, ultrasound.

cholecystitis during the pandemic. The proportion of patients who had early surgery remained the same over time, but some months had a higher proportion of early surgery than previous years, despite patients diagnosed during the pandemic being more comorbid. This might have been a result of knowing that resources for delayed surgery were scarce, as shown in this paper and previous research from Gallriks.[10] The proportion of patients having early surgery was low in both periods, but not lower than previous Swedish evaluations.[21] The practice in Sweden has been restricted on early surgery, limiting this to mildly comorbid patients and those presenting within 5 days of symptoms. Due to concerns for aero-solised virus in asymptomatic patients, open surgery was recommended initially[22]; however, no difference in the rate of open surgery was seen between the two periods.

The differences in ASA classification between the periods could be from otherwise healthy patients staying at home, thinking they had COVID-19. Testing capacity was limited at the beginning of 2020; thus, abdominal pain and fever might have been considered COVID-19 instead of cholecystitis. Chest CT was sometimes performed instead of a nasopharyngeal swab which could explain the higher proportion of CT use.[23] Limited testing may also explain the low number of patients with concurrent COVID-19 infection. Another reason might be that

ultrasound (US) was avoided to prevent close contact and transmission of infection. A third reason for the increase in CT use could be the limited availability of hospital beds, which could have led to CT being used as a tool to decide which patients needed to be admitted instead of waiting for further tests and US.

Comorbid patients might be more likely to seek out healthcare, especially if they fear that they have contracted a potentially deadly infection. This is similar to what the CHOLECOVID collaboration found that patients were sicker during the pandemic.[7] Another possibility is that the documentation of comorbidities got better during the pandemic. However, the CCI was unchanged, suggesting that it was diseases not included in the CCI that led to ASA1 patients being classified as ASA2, for example, hypertension or smoking. Two studies have compared the documentation of code status before and during the pandemic finding that it improved in a UK hospital from a low level and found no change in a Dutch hospital which was already at a high level of documentation.[24 25] However, the documentation of comorbidities has not been investigated in any study.

There was no increase in the grade of cholecystitis and the patients who did seek care did not do so later and were admitted and operated on in a similar time frame as the previous 3 years, which was not investigated by CHOLECOVID or the national Swedish study.[7 10] There

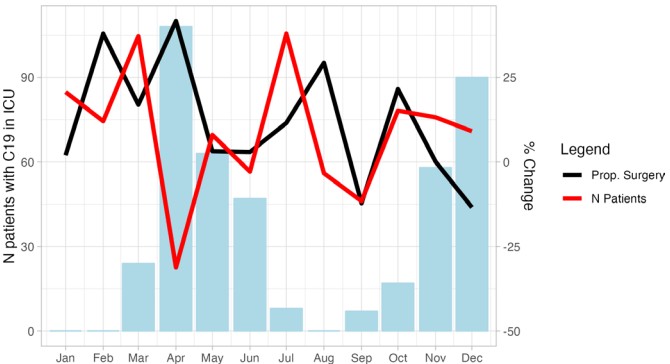

**Figure 1** Number of patients admitted to the intensive care units (ICU) with COVID-19 at the three hospitals, the changes in total cases of cholecystitis and the proportion treated with early surgery during the pandemic compared with 2017–2019.

**Table 3** Logistic regression for delayed surgery within 6 months

| Characteristic | OR | 95% CI | P value |
|---|---|---|---|
| Year | | | |
| 2017–2019 | — | — | |
| 2020 | 0.47 | 0.29 to 0.74 | 0.002 |
| ASA | | | |
| 1 | — | — | |
| 2 | 0.72 | 0.45 to 1.17 | 0.2 |
| 3 | 0.22 | 0.13 to 0.38 | <0.001 |
| 4 | 0.05 | 0.01 to 0.16 | <0.001 |

ASA, American Society of Anesthesiologists.

were no other indications that patients presented later, for example, the leucocyte count or CRP.

Emergency surgical care for patients with cholecystitis does not seem to have been affected by the pandemic in Sweden. If this is the result of universal healthcare in contrast with the USA or from more relaxed policies in contrast with Germany and Canada is not entirely clear.[11 12] The incidence and proportion of operatively managed patients did not decrease in Canada, which has universal healthcare but had tougher policies initially. In the USA, there was a demographic shift towards younger and privately insured patients compared with before the pandemic.[11] In Germany, with the strictest policies of all, the incidence of cholecystitis rose.[8] However, it is not entirely comparable to Sweden since it runs an insurance-based healthcare system with an abundance of hospital beds.[26]

The follow-up and subsequent delayed surgery for NOM patients were further delayed by the pandemic. This is in line with guidelines at the time prioritising emergency surgery and malignancy, and previous research findings.[2 10] The median wait time for delayed cholecystectomy was too long before the pandemic and the limited resources of the pandemic increased it by almost 2 months. Despite this, there was no increase in recurrent gallstone complications when adjusting for follow-up of patients diagnosed during the pandemic. Previous research in the same population highlighted that recurrent gallstone disease is most common in the first months after discharge and common in NOM patients.[16] Thus, delayed surgery did not stop them previously, and delaying it further did not change this.

## Strengths and limitations

One of the main strengths of the study is the inclusion of all patients, not only those undergoing surgery. However, patients cared for by general practitioners would be missing from the analysis. Based on our clinical experience, most of these patients were previously referred to the hospital if they had a fever and abdominal pain making this less of an issue. However, they may not have been referred during the pandemic if they were otherwise healthy or thought to have abdominal symptoms from COVID-19.

This study included patients throughout the entirety of 2020 compared with CHOLECOVID, which only included patients from the first 2 months of the pandemic. The risk of selection bias is lower than in a prospective study though with the drawbacks of a retrospective study where inclusion was dependent on correct classification in the EPR. Standardisation of data collection was attempted and by using EPR data recall bias is limited. However, some notes lacked detail. The data are likely representative of countries with similar healthcare systems and pandemic responses. Sensitivity analysis gave a required effect size of 0.22 for the Wilcoxon ranked sum test to detect a difference at 95% power and $\alpha=0.05$ and 0.1 for the $\chi^2$ test to detect a difference at 4 df. The logistic regression was

sufficiently powered, while the Cox proportional hazards regression was powered to detect an increase of 50% in complications, thus smaller differences are not ruled out.

The study only investigated the treatment for the first cholecystitis in the period 2017–2020 and subsequent events were classified as new gallstone complications. Patients with their first cholecystitis at the end of 2019 would also be impacted by the pandemic and a reduced capacity for elective surgery in 2020. The subsequent waves in 2021 were not included in this study; thus, the study is limited to the initial response and changed policies. The pandemic group does not include the first 2 months of the year, where fewer patients have cholecystitis due to seasonal variation. The comparisons with previous years were performed on a month-to-month basis and the incidence was adjusted for season to account for this.[27]

## Future perspectives

This study investigated the first cholecystitis in patients during the period 2017–2020 and did not find a change in the incidence. While there was no increased incidence of cholecystitis in the present study, other emergency general surgery diseases were affected by the pandemic, for example, the incidence of appendicitis increased according to Swedish studies, potentially as a direct effect of COVID-19 infection.[28] The national patient registry which contains data on diagnoses and surgical treatment could allow for the evaluation of a national material and cover other emergency general surgery diseases besides cholecystitis. It would be especially interesting to investigate emergency general surgery diseases that are usually treated without surgery or antibiotics, for example, diverticulitis, to assess if patients stayed away from the hospitals during the pandemic.

## Conclusion

At our centres, emergency surgery for cholecystitis was not impacted by the pandemic. Patients were more comorbid but did not have a more severe grade of cholecystitis nor did they seek care later. Fewer patients NOM had elective surgery within 6 months of their initial diagnosis but there was no corresponding increase in gallstone complications.

**Acknowledgements** We want to thank the EPR and analysis departments at Region Gävleborg and Region Uppsala for their help with identifying the population from ICD codes. We also want to thank the surgeons and administrators reporting to Gallriks for making the study possible. Thanks to Anders Blomberg and Jennie Wickenberg for discussions regarding the design of the study. Thanks to Louise Helenius and Tamali Majumder for helping with the collection of data.

**Contributors** EO and FL substantially contributed to the conception and design of the study. EO, SJ and CL acquired the data with the support of FL. EO and SJ performed analysis and interpretation of data. EO, SJ and FL drafted the article. All authors revised it critically for important intellectual content. All authors approved the final version to be published. EO is responsible for the overall content as guarantor.

**Funding** The research was funded by the Centre for Research and Development, Gävleborg Region grant numbers CFUG-965514, CFUG-965517. EO and CL are funded by Region Gävleborg, no grant number. FL was financed by grants from the

Swedish state under the agreement between the Swedish Government and the Uppsala County Council (ALF), no grant number. Open access funding from Uppsala University, no grant number.

**Competing interests** None declared.

**Patient and public involvement** Patients and/or the public were not involved in the design, or conduct, or reporting, or dissemination plans of this research.

**Patient consent for publication** Not applicable.

**Ethics approval** The Swedish Ethical Review Authority approved the study, number 2021-00862. Patients with data from Gallriks had consented to be included in the registry for research purposes, for patients not included in Gallriks or treated NOM, the Swedish Ethical Review Authority waived the requirement of consent since the study was retrospective and without intervention or risk to the patients.

**Provenance and peer review** Not commissioned; externally peer reviewed.

**Data availability statement** Data are available on reasonable request. Data may be obtained from a third party and are not publicly available. The data are not publicly available due to information that could compromise the privacy of research participants. The registry data that support the findings of this study are available from Gallriks (https://www.ucr.uu.se/gallriks/). Restrictions apply to the availability of these data why the authors cannot share them. EPR data are, however, available from the corresponding author on reasonable request.

**ORCID iD**
Erik Osterman http://orcid.org/0000-0003-1621-7872

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
