## [Reviewer comments · BMJ Open]

ARTICLE DETAILS

TITLE (PROVISIONAL)	Effect of the COVID-19 Pandemic on the Care for Acute Cholecystitis – a Swedish Multicenter Retrospective Cohort Study
AUTHORS	Osterman, Erik; Jakobsson, Sofia; Larsson, Christina; Linder, Fredrik

VERSION 1 – REVIEW

REVIEWER	Gonzalez-Castillo, Ana Universitat Pompeu Fabra, General Surgery
REVIEW RETURNED	14-Sep-2023

GENERAL COMMENTS	Congratulations for the work and results! It is an excellent resource management, and some Countries should learn from you in the next pandemic. I just wanted to make some suggestions for improvement: In methods, did you assessed the quantitative variable with using the Kolmogorov-Smirnov test) If it is yes, I recommend you describe it in methods. There is only a 50% of surgical treatment and there is no discussion about this low percentage of surgical treatment. You should justify why the percentage is low, even in Grade 1 from TG without pandemic restrictions. In line 42 in Conclusions, I suggest adding: “was not impacted by the pandemic in Sweden”. In line 49 I suggest adding “in a developed country with few restrictions”. In line 66 the paragraph “Due to concerns over of aerosolized virus in asymptomatic patients open surgery was preferred initially. This might have resulted in fewer emergency cholecystectomies, as postoperative complications are more common, especially in obese patients. In combination with reduced capacity for elective cholecystectomies, the concern was a possible increase in the number of recurrent gallstone complications”. From my point of view, I think it should be described in the discussion. For the background is correct to end in line 64. I recommend you read this article and I suggest you to add it in the background and discussion. They show interesting results that are the opposite of yours.
---

	https://wjeb.biomedcentral.com/articles/10.1186/s13017-022-00466-4
REVIEWER	Schroeppel, Thomas University of Colorado Colorado Springs
REVIEW RETURNED	27-Sep-2023
GENERAL COMMENTS	Nice revisions. I think this manuscript is more appropriate for the wider audience of BMJ Open as opposed to TSACO. No further queries.

VERSION 1 – AUTHOR RESPONSE

Reviewer: 1

Dr. Ana Gonzalez-Castillo, Universitat Pompeu Fabra

Comments to the Author:

Congratulations for the work and results! It is an excellent resource management, and some Countries should learn from you in the next pandemic.

- Thank you for your kind comments on the manuscript, we have done some edits based on your suggestions, see below!

I just wanted to make some suggestions for improvement:

In methods, did you assessed the quantitative variable with using the Kolmogorov-Smirnov test) If it is yes, I recommend you describe it in methods.

- All numeric/continuous variables were assumed non-parametric. Thus no KS tests were performed.

There is only a 50% of surgical treatment and there is no discussion about this low percentage of surgical treatment. You should justify why the percentage is low, even in Grade 1 from TG without pandemic restrictions.

- We added two sentences in the first paragraph of the discussion. We agree it is low, but it is what it is, we are confident that we have identified all patients who had surgery at our three hospitals.

-

“The proportion of patients having early surgery was low in both periods, but not lower than previous Swedish evaluations. The practice in Sweden has been restrictive on early surgery, limiting this to mildly comorbid patients and those presenting within 5 days of symptoms.”

In line 42 in Conclusions, I suggest adding: “was not impacted by the pandemic in Sweden”.

- It has been added.

In line 49 I suggest adding “in a developed country with few restrictions”.

- Ok, added to the Strengths and limitations

In line 66 the paragraph “Due to concerns over of aerosolized virus in asymptomatic patients open surgery was preferred initially. This might have resulted in fewer emergency cholecystectomies, as

postoperative complications are more common, especially in obese patients. In combination with reduced capacity for elective cholecystectomies, the concern was a possible increase in the number of recurrent gallstone complications". From my point of view, I think it should be described in the discussion. For the background is correct to end in line 64.

- The paragraph has been revised and moved to the discussion. It now follows the first paragraph and reads:

"Due to concerns for aerosolized virus in asymptomatic patients open surgery was recommended initially,[7] however no difference in the rate of open surgery was seen between the two periods."

I recommend you read this article and I suggest you to add it in the background and discussion. They show interesting results that are the opposite of yours.

<https://wjes.biomedcentral.com/articles/10.1186/s13017-022-00466-4>

- Thank you for the suggestion, the above study investigates the differences between COVID19 infected and non-infected patients and not between non-pandemic and pandemic time periods. They reference other articles highlighting a potential increase in the number of gangrenous cholecystitis, but none are larger than Cholecovid which shows the same finding and is referenced. We had only a few patients with concomitant COVID-19 in the present study making the above study less relevant for our findings.

Reviewer: 2

Dr. Thomas Schroepel, University of Colorado Colorado Springs

Comments to the Author:

Nice revisions. I think this manuscript is more appropriate for the wider audience of BMJ Open as opposed to TSACO. No further queries.

- Thank you!